# Comirnaty-Elicited and Convalescent Sera Recognize Different Spike Epitopes

**DOI:** 10.3390/vaccines9121419

**Published:** 2021-12-01

**Authors:** Sascha Hein, Nuka Ivalu Benz, Jonathan Eisert, Marie-Luise Herrlein, Doris Oberle, Michael Dreher, Julia C. Stingl, Christoph Hildt, Eberhard Hildt

**Affiliations:** 1Department of Virology, Paul-Ehrlich-Institut, Paul-Ehrlich Street 51–59, D-63225 Langen, Germany; nuka.benz@pei.de (N.I.B.); jonathan.eisert@pei.de (J.E.); Marie-Luise.Herrlein@pei.de (M.-L.H.); 2Division of Pharmacovigilance, Paul-Ehrlich-Institut, Paul-Ehrlich Street 51–59, D-63325 Langen, Germany; doris.oberle@pei.de; 3Department of Pneumology and Intensive Care Medicine, RWTH Aachen University Hospital Aachen, Pauwelsstraße 30, D-52074 Aachen, Germany; mdreher@ukaachen.de; 4Institute of Clinical Pharmacology, RWTH Aachen University Hospital Aachen, Pauwelsstraße 30, D-52074 Aachen, Germany; jstingl@ukaachen.de; 5Main-Kinzig-Kliniken, Herzbachweg 14, D-63571 Gelnhausen, Germany; Christoph.Hildt@mkkliniken.de

**Keywords:** SARS-CoV-2 spike, linear epitopes, convalescent, peptide array, stabilized spike protein

## Abstract

Many of the approved SARS-CoV-2 vaccines are based on a stabilized variant of the spike protein. This raises the question of whether the immune response against the stabilized spike is identical to the immune response that is elicited by the native spike in the case of a SARS-CoV-2 infection. Using a peptide array-based approach, we analysed the binding of antibodies from Comirnaty-elicited, convalescent, and control sera to the peptides covering the spike protein. A total of 37 linear epitopes were identified. A total of 26 of these epitopes were almost exclusively recognized by the convalescent sera. Mapping these epitopes to the spike structures revealed that most of these 26 epitopes are masked in the pre-fusion structure. In particular, in the conserved central helix, three epitopes that are only exposed in the post-fusion conformation were identified. This indicates a higher spike-specific antibody diversity in convalescent sera. These differences could be relevant for the breadth of spike-specific immune response.

## 1. Introduction

The coronavirus-induced disease 2019 (COVID2019) pandemic is caused by the severe acute respiratory syndrome coronavirus 2 (SARS-CoV-2). By August 2021 (10 September 2021), over 223 million cases of COVID-19 have been reported and more than 4.6 million people have died from this disease (source: COVID-19 Dashboard by the Center for Systems Science and Engineering (CSSE) at Johns Hopkins University (JHU)). To control this pandemic, different vaccines based on the S protein of the virus have been developed. The S protein is a homotrimer, with each monomer consisting of two major domains (S1 and S2) [1]. The S1 subunit is further divided into the N-terminal domain (NTD) and the receptor binding domain (RBD) [2]. The RBD is capable of binding to the ACE2 (angiotensin converting enzyme 2), which serves as a receptor and initiates cell entry. After the RBD binds to ACE2 and after the proteolytic cleavage between the S1 and S2 domains caused by the cellular protease furin, the N-terminal domain (S1) of the protein dissociates. Subsequently, the S2 domain folds back to the postfusion structure and releases the fusion peptide, which binds to the membrane of mammalian cells and is responsible for virus uptake. The S protein can be cleaved by furin not only after binding to ACE2, but it can also be cleaved during virus replication, when the structural proteins are transferred to the cell membrane [3]. To circumvent this labile structure of the S protein, many vaccines based on various platforms such as mRNA in the case of Comirnaty (BioNTech/Pfizer) and mRNA-1273 (Moderna), an adenoviral vector in case of Ad26.COV2.S (Janssen-Johnson & Johnson), or a purified recombinant S protein in case of NVX-CoV2373 (Novavax)) have been designed using a pre-fusion stabilized form of the S protein [4,5]. To create this structure, two mutations K986P and V987P are inserted at the beginning of the central helix of the spike protein [6]. This stabilized form is characterized by reduced flexibility and is resistant against proteolysis by host proteases [7]. Moreover, these mutations prevent processing and thereby the rearrangement of the S2 domain [6,7]. These differences could play a major role, especially for the recognition of linear epitopes. A variety of linear epitopes exists within the S protein, as evidenced by the analysis of convalescent sera [8,9,10,11]. This raises the question of whether, in case of vaccination with a construct stabilized in the pre-fusion conformation, predominantly linear epitopes are recognized, whereas in convalescent individuals, whether the epitopes localized in regions that are masked in the pre-fusion structure may be recognized. To address this question, a comparative epitope mapping of vaccine (Comirnaty)-elicited sera and convalescent sera was performed.

## 2. Materials and Methods

The study (PEI-SARS-CoV-2) was performed according to the principles of the Declaration of Helsinki. It was approved by the local ethics committee (Ethik-Kommission, Landesärztekammer Hessen, 60314 Frankfurt am Main, Germany). A written declaration of consent was obtained from all participants (2020-1664_3-evBO). The studies with the convalescent sera provided by the RWTH cBMB were performed in accordance with the standing orders of the cBMB and in accordance with the approval granted by the ethics committee (Ethikvotum 206/09 der Ethikkommission der Medizinischen Fakultät der RWTH Aachen University). In the COVID-19 Aachen Study COVAS, n = 143 samples from n = 72 acute COVID-19 patients were collected during the first wave of infections in the period of March until August 2020 at different time points during hospital stay and during ambulant follow-up hospital visits. From the samples stored in the COVAS biobank at the university hospital Aachen, n = 31 patient serum samples have been selected for this analysis. A total of N = 14 patient serum samples were obtained during the first week of hospitalization, and n = 17 samples were from patients more than n = 50 days after hospitalization. A total of N = 13 patients had acute respiratory distress syndrome, and n = 18 patients were hospitalized with a milder course of illness (non-ARDS). The mean age of the patients with ARDS was 61 (+/−10) years of age, and the mean age of the the non-ARDS patients was 64 (+/−19). Nine of the total samples (n = 31) were derived from female patients and n = 22 were derived from male patients.

### 2.1. Anti-RBD ELISA

Spike-RBD-specific antibodies were quantified by ELISA. Soluble RBD protein was produced in HEK293T cells and purified by Ni-NTA affinity chromatography according to the protocol from Amanat et al., 2020 [12]. Florian Kramer, Icahn School of Medicine at Mount Sinai, kindly provided the expression plasmid pCAGGS-sRBD. The coating of 96-well microtiter plates (costar 3590, Corning Incorporated, Kennebunk, ME, USA) was perfomed with 50 µL (4 µg/mL RBD in PBS) over night at 4 °C. Plates were blocked for 1 h at RT with 10% FCS in PBS and were washed between each step three times with PBS-T (PBS supplemented with 0.05% Tween). The sera were prediluted 1:100 in 10% FCS in PBS and 50 µL per well and were incubated for 1.5 h at RT. As a secondary antibody, 50 µL of anti-human IgG HRP-linked ECL antibody (Cytiva, Dassel, Germany) was used in a 1:3000 dilution. After 1 h of incubation at RT, the ELISA plates were developed with 75 µL TMB substrate solution (eBioscience, San Diego, USA) for 5 min, stopped with 75 µL 1N sulfuric acid, and analyzed directly at 450 nm on a Tecan reader (Tecan Group, Männedorf, Switzerland).

### 2.2. Synthesis of Peptides and Preparation of Peptide Arrays

To analyze the linear epitopes of spike-specific antibodies, which are elicited either by vaccination with Comirnaty or by natural infection (from the first wave), a total of 253 15-mer peptides (offset of 5 amino acids) covering the complete primary sequence of the viral spike protein were synthesized according to the principles of Merrifield [13]. Using a slide-spotting robot (Intavis AG, Tuebingen, Germany), the peptides were immobilized in a duplicate on a cellulose membrane (Intavis AG, Tuebingen Germany) attached to a 76 mm × 26 mm slide. Membranes with spotted peptides were stored at 4 °C until use.

### 2.3. Experimental Design of Peptide Arrays

A total of 60 samples were analyzed, 25 sera from Comirnaty vaccinated patients and 31 from non-vaccinated convalescent patients, and 4 sera served as negative controls (without previous SARS-CoV-2 infection and vaccination). Peptide arrays were performed according to a modified protocol from Kühne et al. 2015 [14]. Peptide arrays were blocked for 2 h at room temperature with Casein blocking buffer (Sigma Aldrich, Hamburg, Germany) diluted 1:10 in Tris-buffered saline (TBS-T, pH 7.4) containing 0.05% Tween-20 and 5% sucrose. Afterwards, each membrane was washed with TBS-T (0.5% Tween-20, pH 7.4) for 5 min at room temperature and was dried for 1 h. Thereafter, peptide arrays were incubated overnight at 4 °C with the sera diluted 1:40 in a total volume of 1.8 mL blocking buffer. Following overnight incubation, the membranes were washed for 5 × 5 min with TBS-T (0.1% Tween-20, pH 7.4) at room temperature. For human IgG detection, the peptide arrays were incubated with the secondary antibody goat anti-human IgG IRDye680 (LI-COR, Bad Homburg, Germany) diluted 1:10000 in blocking buffer at room temperature. Secondary antibody incubation was followed by five 5 min wash steps with TBS-T (0.1% Tween-20, pH 7.4), and the membranes were dried for 1 h. Afterwards, the arrays were scanned on a fluorescence detector LI-COR Odyssey CLx (LI-COR, Bad Homburg, Germany). The scanned arrays were analyzed with the software Image Studio Lite (LI-COR. Bad Homburg, Germany) and the fluorescence intensity values were extracted. For analysis, the log_2_ fluorescence intensity was used.

### 2.4. Statistical Analysis

All of the statistical analyses were performed with R and with the software Prism 8 (GraphPad, San Diego, CA, USA). The standard Z-Score of each group (vaccinated, convalescent sera) was calculated using the following equation:$$Z-\mathrm{Score}:\frac{M_{i}-M_{all}}{\sigma_{all}}$$

The mean of each peptide from the respective group was subtracted by the mean of all of the arrays. For normalization, the values were divided through the over-all standard deviation. For *t*-test analysis, we chose an unpaired Welch’s *t*-test and performed the analysis in Prism.

## 3. Results and Discussion

To identify the linear epitopes recognized by vaccine (Comirnaty)-elicited and by convalescent sera, peptide arrays were used that encompassed 253 spike-specific peptides. The cellulose conjugated peptides were synthesized using the CelluSpot technology and were spotted in duplicate (Appendix A). The IgG responses of sixty sera (25 Comirnaty double vaccinated, 31 convalescent patient sera, 4 control naïve sera) were analyzed with the peptide array (Figure 1A).

By selecting the top 50 peptides with the highest signals in the array and by comparing their cumulative mean values, it can be seen that the vaccine-elicited sera and convalescent sera differ from the control (Figure 2A). A weakly higher but not statistically significant cumulative fluorescence signal was found for the convalescent sera compared to the vaccine-elicited sera. To control whether the higher cumulative fluorescence intensities in the convalescent sera were due to a higher antibody titer, the titer was determined by ELISA (Figure 2B).

The anti-RBD ELISA revealed that the vaccine-elicited sera have a higher titer than the convalescent sera. Several studies have shown that there is a linear correlation between the anti-RBD titer of sera and the anti-spike titer [15]. On the one hand, the titer of the convalescent sera was lower than that of the vaccinated sera, and on the other hand, the cumulative fluorescence intensity is slightly higher in the convalescent sera. These results can be explained by the hypothesis that the convalescent sera recognize more linear epitopes than the vaccine-elicited sera.

For a detailed comparison of the two serum types, the mean values of all of the control sera were subtracted from the values that were obtained for the vaccinated-elicited sera and convalescent sera, respectively, and the Z-Score was calculated for each group (Figure 1B). In total, 37 peptides with a Z-Score above 1.5 (Table 1) were identified. A total of 26 of the peptides with a Z-Score of ≥1.5 were present exclusively in the convalescent sera. A single significant epitope (S-171) was found to be present exclusively in the vaccinated-elicited sera. The other 10 peptides (S-5, S-12, S-18, S-37, S-176, S-198, TM-242, TM-245, TM-246, CT-249) with a Z-score of ≥1.5 were able to be recognized by both sera types. This reflects a clear difference with respect to the linear epitopes recognized by convalescent and Comirnaty-elicited sera. The 10 peptides that were common to both groups are mainly located in two regions of the spike protein. A total of four epitopes (TM-242, TM-245, TM-246, CT-249) were located in the transmembrane/cytoplasmic domain near the C-terminus, and another four epitopes (S-5, S-12, S-18, S-37) were located in the N-terminal domain (NTD). These four peptides all contain highly flexible loop regions of the previously described antigen supersite in the S protein (Figure 3A–C) [16]. Furthermore, the seven peptides (S-50, S-53, S-54, S-56, S-57, S-58, S-60) that have a Z-score of ≥1.5 are only present in the group of convalescent sera. These peptides are also located to the flexible loop region of the NTD. In particular, the binding of monoclonal antibodies was described for regions of peptides S-5 and S-50. Peptide S-12 also contained amino acids H69/V70, which are deleted in SARS-CoV-2 variant cluster 5 and B.1.1.7. This deletion leads to an escape of different monoclonal antibodies [17].

Three other peptides (S-5, S-18 and S-37) near the antigen supersite were detected by the convalescent and the vaccinated-elicited sera. Since these epitopes are masked by the S1 domain of another spike monomer, they are only accessible once the S1 domain is dissociated. Antibodies against these epitopes can therefore be generated when the native spike protein is used as antigen. Interestingly, antibodies against already described linear epitopes were mainly found in significant amounts in the convalescent sera (Figure 3A,C; Table 1). This confirmation of linear epitopes demonstrates the functionality of the peptide array setup.

Two peptides (RBD-99 and RBD-101) harboring the N501 position, which is mutated in B.1.1.7, B.1.351 and P.1 variants were found. A third peptide (RBD-100) is located between these two peptides and has a Z-Score of 1.4 for the vaccine-elicited sera group. This region is in direct contact with ACE2 and is thus only accessible in the open state form of the protein [20]. Possibly, the stabilization of the spike protein favors the formation of the closed state form of the protein and leads to a reduced exposure of this epitope. No linear epitope was detected at the furin cleavage site. However, a very strong signal for the linear epitope (S-133: ECDIPIGAGICASYQ; Z-score: 2.7) is located at the furin-protease flanking region site (QTQTN), which is exclusively recognized by convalescent sera and is only exposed when furin has cleaved the S1 domain. This and eight further peptides (Table 1; RBD-75, RBD-76, RBD-106, RBD-107, S-109, S-124, S-125, S-127) are accessible in the cleaved S1 domain. The antibodies binding to this region prevent the spread of the free S1 domain in the body. Another very interesting peptide that is found in both serum types is the S-198: KVEAEVQIDRLITGR. This contains the amino acids K986 and V987, which are both exchanged against proline in the pre-fusion stabilized spike protein of the Comirnaty as well as in the mRNA-1273 vaccine. By looking at the structure of the spike protein, it is noticeable that part KVEAEVQIDR of the peptide is accessible in the opened-state structure and can thus function as a linear epitope in both cases (Figure 3). In the case of the convalescent sera, six peptides (S-199, S-200, S-201, S-202, S-203, S-204) with a Z-score of more than 1.5 were found in the direct vicinity. These six peptides are part of the central helix (CH) in the spike protein that is in direct contact with all of the monomers of the spike and to which there is no access in the pre-fusion state (Figure 3B). It is notable that parts of the first three peptides (S-199–S-201; Table 1) were identical to the epitope TGRLQSLQTYVT that was previously detected in 15% of all of the convalescent sera [8]. The other three peptides (S-202, S-203, S-204) cover parts of the RASANLAATKMSECVLG linear epitope [19]. The regions of this helix are enabled after the dissociation of the S1 domain and the rearrangement of the structure into the post-fusion structure (Figure 3D) [21]. In addition, the N-terminal part of this post-fusion structure has a large linker region, making the entire CH accessible to antibodies. Studies have revealed that up to 70% of the spike proteins on virus particles as well as on infected cells are in the post-fusion conformation [22]. Other studies have shown that the S protein interacts with the structural proteins M and E during the maturation of the spike protein and in the virus particle [23,24]. In vaccine-induced spike protein production, these interactions do not occur. This could have an impact on post-translational N-glycan modifications. The difference in the glycosylation pattern, in addition to the stabilizing mutations K986P and V987P, could play a critical role in S protein flexibility and linear epitope recognition.

The immune response to the spike might also be affected by the presence of further structural proteins interacting with the spike, as in case of the natural infection. In contrast to this, the spike produced after vaccination with Comirnaty or other vector-based vaccines is exclusively produced in the absence of further viral structural proteins. This could have an impact on the structure of spike as well as on the processing and the accessibility of the spike epitopes.

Furthermore, we evaluated the data of the convalescent sera in relation to the non-ARDS (n = 18) and ARDS (n = 13) groups (Appendix A). As already described by Zhang et al., we could identify more epitopes in severe ARDS cases than we could in the non-ARDS with a Z-score of ≥1.5 (Appendix A) [19]. No epitope with a Z-score above 1.5 was only identified exclusively in the non-ARDS sera. The 23 identified peptides from Table 1 (S-12, S-18, S-37, S-56, S-57, S-60, RBD-75, RBD-76, RBD-99, RBD-107, S-124, S-125, S-127, S-133, S-139, S-171, S-176, S-197, S-198, S-201, S-204, TM-246, CT-249) are found in both convalescent groups ARDS and non ARDS. The other 14 peptides (S-5, S-50, S-53, S-54, S-58, RBD-101, RBD-106, S-109, S-199, S-200, S-202, S-203, TM-242, TM-245) were identified exclusively in ARDS patients with a Z score of ≥1.5. However, for a concrete comparison of the ARDS and non-ARDS groups, a larger number of samples is needed.

This study clearly shows that Comirnaty-elicited sera trigger the linear epitopes in the case of a different B-cell response as opposed to a native spike protein in the context of SARS-CoV-2. This study reveals that in convalescent sera, a significant higher number of linear epitopes is recognized. Although vaccination with a pre-fusion stabilized spike protein results in a strong immune response against S1 and S2, in particular, the regions of the central helix are less recognized as epitopes. In addition, based on these results, it can be speculated that the diversity of the antibodies produced by Comirnaty-elicited sera is lower than it is in convalescent sera although the antibody titer is higher in Comirnaty-elicited sera compared to convalescent sera. This observation is independent from the outcome of infection (non-ARDS or ARDS). Several publications have shown that antibody binding to the RBD, the NTD, and the S2 domain can be neutralizing, whereas the efficacy of the individual domains varies [25]. It has been described that the antibodies that recognize the RBD contribute more than 90% to neutralization capacity of the sera [4,26]. However, a limitation of neutralization assays is their design. The most common tests are based on a virus-like particle neutralization assay, which indicates the extent to which the antibodies in a serum can prevent the cell entry of virus-like particles that present the S protein on their surface [27,28]. In other words, it interacts with the S-protein on the surface of the virus-like particle, but not with the S protein on the surface of the infected cell that is not present under these experimental conditions. However, whether infected cells that have spike proteins in the post fusion structure on their surface are recognized by these antibodies is rarely studied [29]. The elimination of infected cells is particularly important in long-persisting infections with low viral titers; otherwise, they may fuse with neighboring cells using the post fusion spike protein [30,31]. In such cases, neutralizing antibodies against the CH could play an important role. The role of CH as potential neutralization site will be a part of our further research. The linear epitopes of the CH are not altered in the currently circulating SARS-CoV-2 variants compared to the wild type. It seems that this area of the spike protein is less frequently affected by mutations than the RBD or the NTD. The antibodies targeting these regions could thus provide additional protection against emerging SARS-CoV-2 variants. Due to the ongoing mutation of SARS-CoV-2, the proportion of neutralizing antibodies decreases; thus, each neutralizing epitope increases in importance. If antibodies against CH also have a strong neutralizing effect, then the development of therapeutic antibodies against CH could be an important step in the fight against SARS-CoV-2 and its mutants.

## 4. Conclusions

In this study, we found that the sera derived from Comirnaty-vaccinated individuals recognize fewer linear epitopes than sera derived from convalescent patients. This is the case although the antibody titer of the Comirnaty-elicited sera is higher than the titer of the sera from convalescent patients. The reason for this difference could be the stabilization of the spike protein. If this is the case, then the number of linear epitopes and thus the diversity of antibodies in the sera should be significantly reduced in all vaccines that use a stabilized spike protein as the antigen. This could have an impact on the breadth of the immune response, which could be relevant for protection from variants harbouring escape mutations within the spike. This study demonstrates the importance of vaccine design in eliciting an immune response that is equivalent to that of the pathogen.

## Figures and Tables

**Figure 1 vaccines-09-01419-f001:**
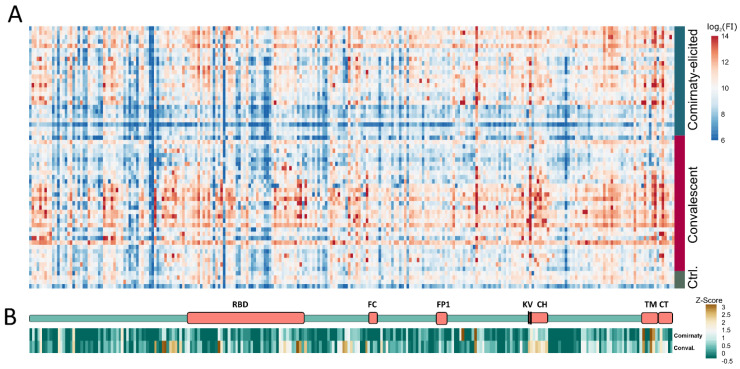
Mapping of linear S protein epitopes. (**A**) Heatmap of IgG antibody immune response of 31 sera from COVID-19 convalescent patients, 25 BNT162b2 vaccinated patients, and 4 healthy donors. FI: fluorescence intensity. (**B**) Schematic structure of the S protein and condensed heatmap of all 56 sera grouped in vaccinated and convalescent patients. RBD: Receptor binding domain; FC: furin cleavage site; FP: fusion peptide; KV: amino acid positions K986 and V987, which are mutated to proline in the pre-fusion stabilized S protein; CH: central helix; TM: transmembrane domain; CT: cytoplasmic tail.

**Figure 2 vaccines-09-01419-f002:**
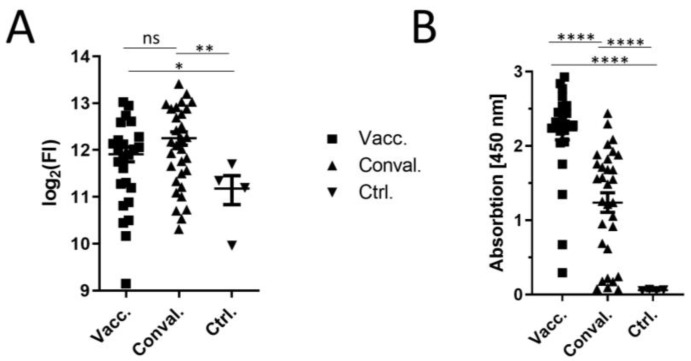
Validation of the peptide array results. (**A**) FI means of the 50 highest peptides from the different sera groups. (**B**) Anti-RBD ELISA of the different sera groups. Vacc.: vaccine-elicited sera; Conval.: convalescent; Ctrl.: Neg. sera. (non-vaccinated and non-infected) * *p*-value ≤ 0.05; ** *p*-value ≤ 0.01; **** *p*-value ≤ 0.0001.

**Figure 3 vaccines-09-01419-f003:**
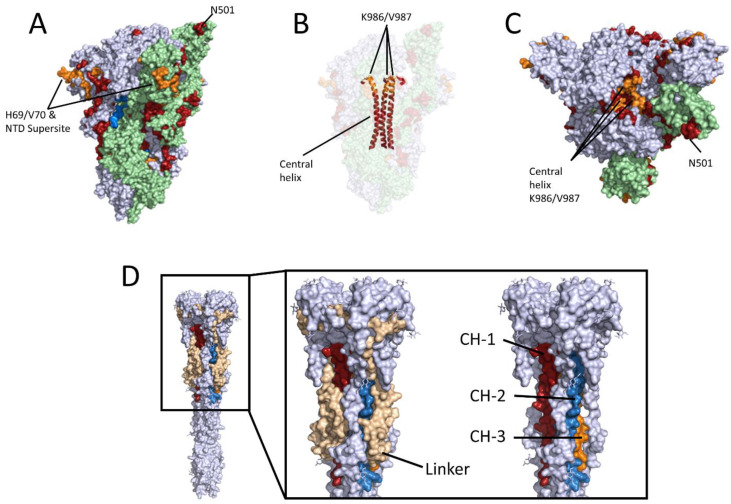
Location of the linear epitopes in the pre-fusion stabilized S protein and post-fusion S protein structure. (**A**–**C**) Pre-fusion stabilized S Protein (PDB: 6VYB). Gray: pre-fusion dimer in closed-state; green: pre-fusion monomer in open-state (RBD-UP); blue: linear epitopes present only in vaccinated sera; orange: linear epitopes present in both groups; red: linear epitopes exclusive in convalescent sera. (**D**) Post-fusion S protein (PDB: 6M3W) exposed the CH helices of all three monomers. Gray: Post-fusion trimer; red/blue/orange: CH of the three monomers; beige: linker region.

**Table 1 vaccines-09-01419-t001:** Identified linear S protein epitopes in vaccinated-elicited sera and convalescent sera. Red: Z-score ≥ 1.5. Highlighted: Blue: epitopes significant in both sera groups; red: epitopes only significant in convalescent sera; green: epitopes only significant in vaccinated-elicited sera.

Location-Peptide Nr.	Z-Score vacc. sera	Z-Score con. val. sera	Peptide ^1^	Conspicuity	Previously Described ^2^
S-5	**1.5**	**1.8**	RTQLPPAYTNSFTRG	**Neutralization supersite**	
S-12	**1.7**	**1.9**	LPFFSNVTWFHAI**HV**		
S-18	**1.9**	**2.1**	FNDGVYFASTEKSNI		**FDNPVLPFNDGVYFA** [18]
S-37	**1.8**	**1.5**	GKQGNFKNLREFVFK		
S-50	−0.9	**2.5**	RSYLTPGDSSSGWTA	**Neutralization supersite**	
S-53	0.8	**3.0**	GAAAYYVGYLQPRTF		
S-54	0.8	**3.4**	YVGYLQPRTFLLKYN		
S-56	0.0	**2.1**	LLKYNENGTITDAVD		
S-57	0.4	**2.4**	ENGTITDAVDCALDP		
S-58	0.0	**2.2**	TDAVDCALDPLSETK		
S-60	0.7	**2.0**	LSETKCTLKSFTVEK		
RBD-75	1.2	**3.1**	SASFSTFKCYGVSPT		**TFKCYGVSPT**KLNDL [11]
RBD-76	0.5	**1.8**	TFKCYGVSPTKLNDL		**TFKCYGVSPTKLNDL** [11]
RBD-99	1.4	**1.5**	PLQSYGFQPT**N**GVGY	**N501 loop**	
RBD-101	0.2	**2.0**	**N**GVGYQPYRVVVLSF		
RBD-106	0.3	**3.2**	GPKKSTNLVKNKCVN		
RBD-107	−0.7	**2.6**	TNLVKNKCVNFNFNG		
S-109	0.0	**2.3**	FNFNGLTGTGVLTES		
S-124	−1.0	**2.7**	NCTEVPVAIHADQLT		
S-125	1.0	**2.4**	PVAIHADQLTPTWRV		
S-127	0.4	**2.1**	PTWRVYSTGSNVFQT		
S-133	−0.1	**2.7**	ECDIPIGAGICASYQ	**Close to Furin-cleavage site**	**ECDIPIGAGICASYQ** [10] **CASYQ**TQTNSPRRAR [18] **CASYQ**TQTNSPRRARSV [9] **ECDIPIGAGICA** [8]
S-139	1.0	**1.9**	SIIAYTMSLGAENSV		ARSVA**SQSIIAYTMSLGAENSV**A [9]
S-171	**3.8**	0.0	CAQKFNGLTVLPPLL		
S-176	**3.6**	**2.7**	ALLAGTITSGWTFGA		
S-197	0.3	**2.3**	LSRLD**KV**EAEVQIDR	**Central Helix**	
S-198	**1.7**	**1.9**	**KV**EAEVQIDRLITGR		**TGR**LQSLQTYVT (15,7% Response frequency Sera) [8]
S-199	0.8	**1.7**	VQIDRLITGRLQSLQ		**TGRLQSLQ**TYVT [8]
S-200	1.4	**2.3**	LITGRLQSLQTYVTQ		**TGRLQSLQTYVT** [8]
S-201	0.4	**2.1**	LQSLQTYVTQQLIRA		TGR**LQSLQTYVT** [8]
S-202	0.5	**1.6**	TYVTQQLIRAAEIRA		**RA**SANLAATKMSECVLG [19]
S-203	0.4	**1.7**	QLIRAAEIRASANLA		**RASANLA**ATKMSECVLG [19]
S-204	0.7	**2.2**	AEIRASANLAATKMS		**RASANLAATKMS**ECVLG [19]
TM-242	**3.2**	**2.6**	YEQYIKWPWYIWLGF		
TM-245	**3.5**	**2.8**	IAGLIAIVMVTIMLC		
TM-246	**2.6**	**1.8**	AIVMVTIMLCCMTSC		
CT-249	**1.7**	**1.9**	CSCLKGCCSCGSCCK		

^1^ Bold red-marked amino acids: HV: Deleted in cluster 5 and B.1.1.7 variant; N: N501 is mutated in variant B.1.1.7 and B.1.351 to Y. KV: mutated in Comirnaty derived spike protein to PP. ^2^ Bold marked amino acids: identical to identified peptide.

## Data Availability

All supporting data are included in this manuscript.

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
