# Peer review of "Comirnaty-Elicited and Convalescent Sera Recognize Different Spike Epitopes"

_vaccines, 2021, doi:10.3390/vaccines9121419_

Round 1

Reviewer 1 Report

This study compared the immune response of comirnaty-elicited and SARS-CoV-2 infection convalescent sera to the spike protein with a peptide array-based approach. The convalescent sera were detected significant higher number of linear epitopes recognize than vaccine elicited sera. Beside the already published linear peptides, this study further identified peptides which recognized by vaccine elicited and/or convalescent sera. This work might be useful for COVID-19 vaccine and antibody development.

  1. The author used lower lowercase letters in the figures but uppercase in the manuscript. Please use uppercase in the figures.
  2. While many people do not have reactions after vaccination or SARS-COVID-2 infection. Elisa to confirm the binding of convalescent and vaccine-elicited sera to SARS-COVID-2 S protein are need. No binding of negative control to S protein also need to be confirmed.
  3. Figure 2B: The peptide arrays experiments were based on complete primary sequence of the viral spike protein. After vaccination or SARS-COVID-2 infection, antibodies binding to different epitopes of S protein will present in the sera. RBD is only small part of S protein, to compare the sera titer, anti-S protein or S trimer is more reliable than anti-RBD. Is there any publication shown that the anti-RBD antibody can indicate whole anti-S protein antibody titer? Or do you have the anti-S protein data?
  4. Line 143-146: “27 of these peptides are exclusively present in the convalescent sera, for only one peptide a significant increase in the group of the vaccinated sera was detected and 10 peptides can be are recognized by both sera types.” This sentence is hard to understand.

According to table 1, the 27 peptides are exclusively present in the convalescent sera but not in vaccinated sera. Please clarify.

 In table 1: S-50 to RBD-76, RBD-101 to S-139, S-197, S-199 to S-204 only 25 peptides in total.   Are RBD-99 and RBD-100 also included in the 27 peptides?

The only one peptide is the S-171? Please add into the manuscript)

Please mark or list the 10 peptides can be recognized by both sera types.

  1. In Table 1, RBD-100 peptide’s Z score only 0.7 and 1.4, Lower than 1.5. please explain why you listed this peptide in the table
  2. Line 152. Pepetide 1,2 4? I can’t find these three peptides in the manuscript and in citation 13. Are these three Peptides means S-5, S-12 and S-37? please clarify these three peptides.
  3. Line 153: “Peptide 2 also contains amino acids H96/V70”

Are you sure it’s H96 not H69?

  1. Line 158: “Three other peptides (S-5, S-18 and S-37) near the antigen supersite were only detected by the convalescent sera”

in table 1, the Z-score of these three peptides is >1.5 in both vaccinated sera and convalescent sera. So, these peptides should be recognized by both sera, right? Please correct it.

  1. Line 178: please indicate the 14 peptides
  2. Line 186: please indicate the 6 peptides
  3. Line 191: “The other three peptides (S-201 – S-202) cover parts of the RASANLAATKM”

S-201 – S-202 only listed two peptides, according to table 1 the three peptides should be S-202 to S-204. Please correct it.

Author Response

Response to Reviewer #1:

Comment 1: The author used lower lowercase letters in the figures but uppercase in the manuscript. Please use uppercase in the figures.

Reply 1: We changed all lowercase letters in the figures to uppercase.

Comment 2: While many people do not have reactions after vaccination or SARS-COVID-2 infection. Elisa to confirm the binding of convalescent and vaccine-elicited sera to SARS-COVID-2 S protein are need. No binding of negative control to S protein also need to be confirmed.

Reply 2: In the revised version, we clarified this point. In Fig. 2B we show the binding to of sera to the SARS-CoV-2 RBD. No binding is observed in case of the sera of non-vaccinated / non infected patients that were used as negative control. These sera of the negative control were collected before the pandemic. (P4, L152-156)

Comment 3: Figure 2B: The peptide arrays experiments were based on complete primary sequence of the viral spike protein. After vaccination or SARS-COVID-2 infection, antibodies binding to different epitopes of S protein will present in the sera. RBD is only small part of S protein, to compare the sera titer, anti-S protein or S trimer is more reliable than anti-RBD. Is there any publication shown that the anti-RBD antibody can indicate whole anti-S protein antibody titer? Or do you have the anti-S protein data?

Reply 3: We use a semi-quantitative anti-RBD ELISA to determine the relative amount of anti-SARS-COV-2 antibodies in each sera. Several studies have shown a linear correlation between anti-RBD antibody titer and titer against full trimeric spike protein. For example, see Fig. 2A  in Becker et al., Nature Communications 2021. (https://www.nature.com/articles/s41467-021-20973-3.pdf)

Comment 4: Line 143-146: “27 of these peptides are exclusively present in the convalescent sera, for only one peptide a significant increase in the group of the vaccinated sera was detected and 10 peptides can be are recognized by both sera types.” This sentence is hard to understand.

Reply 4: As requested by the reviewer, we changed this paragraph. (P5 L175-195)

Comment 5: According to table 1, the 27 peptides are exclusively present in the convalescent sera but not in vaccinated sera. Please clarify.

Reply 5: We clarified this paragraph. (P5 L177-195)

Comment 6: In table 1: S-50 to RBD-76, RBD-101 to S-139, S-197, S-199 to S-204 only 25 peptides in total.   Are RBD-99 and RBD-100 also included in the 27 peptides?

Reply 6: We agree with the reviewer, this point was not clear in the manuscript.  We modified this point. A Z-score >= 1.5 was set as the threshold, not >1.5. Peptide RBD-99 is included in the list, but peptide RBD-100 (Z-score 1.4) is not. In the old version, we included peptide RBD-100 because it lies between the other two recognized peptides (RBD-99 and RBD-101), represents the highly exposed N501 region of RBD, and has a Z score close to the cutoff value. In the revised version, we deleted peptide RBD-100 from Table 1 and reduced the 27 peptides to 26 peptides. In addition, a short passage about detection by peptide RBD-100 was included in the text. (P7 L238ff)  

Comment 7: The only one peptide is the S-171? Please add into the manuscript)

Reply 7: As suggested by the reviewer, we added the information about the peptide to the text. (P5 L179ff)   

Comment 8: Please mark or list the 10 peptides can be recognized by both sera types.

Reply 8: According to the comment of the reviewer, we marked these peptides in table 1 in blue and added the information to the text. (P5 L178)

Comment 9: In Table 1, RBD-100 peptide’s Z score only 0.7 and 1.4, Lower than 1.5. please explain why you listed this peptide in the table

Reply 9: This point was addressed in our reply to comment 6. The peptide RBD-100 was deleted from table 1.

Comment 10: Line 152. Pepetide 1,2 4? I can’t find these three peptides in the manuscript and in citation 13. Are these three Peptides means S-5, S-12 and S-37? please clarify these three peptides.

Reply 10: We apologize for this. Indeed, a wrong citation was used. We corrected this and used the right citation.

Comment 11: Line 153: “Peptide 2 also contains amino acids H96/V70” Are you sure it’s H96 not H69?

Reply 11:  Thank you. This was a typo. Yes. It´s H69. We changed this (P5 L194)

Comment 12: Line 158: “Three other peptides (S-5, S-18 and S-37) near the antigen supersite were only detected by the convalescent sera”. In table 1, the Z-score of these three peptides is >1.5 in both vaccinated sera and convalescent sera. So, these peptides should be recognized by both sera, right? Please correct it.

Reply 12: We corrected this paragraph. (P7 L225)

Comment 13: Line 178: please indicate the 14 peptides

Reply 13: As suggested by the reviewer, we modified this paragraph and added the peptide numbers into the text. (P7 L248).

Comment 14: Line 186: please indicate the 6 peptides

Reply 14: We indicate the 6 peptides in the text of the revised manuscript (P8 L256).

Comment 15: Line 191: “The other three peptides (S-201 – S-202) cover parts of the RASANLAATKM” S-201 – S-202 only listed two peptides, according to table 1 the three peptides should be S-202 to S-204. Please correct it.

Reply 15: Thank you. We corrected the numbers to S-202 – S-204. (P8 L256)

Reviewer 2 Report

This manuscript analyzes the ability of anti SARS-CoV-2 antibodies, present in sera from convalescent, Comirnaty-vaccinated and control subjects, to recognize 15-mer linear peptides covering the entire spike protein. The aim is to study the differences, if any, of the immune response elicited by virus (naïve spike protein) and by the vaccine (a stabilized pre-fusion spike protein).

The analysis highlighted that the convalescent sera, among the total 253 peptides, recognized a higher number of epitopes in comparison to the vaccinated sera. Furthermore, most of these peptides encompasses regions of the spike protein that became available upon binding to the ACE receptor only (27). One epitope was exclusively recognized by vaccinated sera and 10 epitopes were recognized by both sera. This outcome could be important in understanding the differences in the immune response elicited by infection and immunization.

Major points:

The number of sera analyzed (31 convalescent, 25 vaccinated sera) is maybe limited for drawing definitive conclusions. It was indicated that the convalescent sera are from subjects who developed either severe or mild forms of the illness and that serum was collected at different time points. However, no indication of these conditions is reported in the further analysis. It would be interesting to identify, for example, the epitopes against which subjects with ARDS have rised antibodies compared to non-ARDS subjects. However, to this intent a higher number of samples should be analyzed.

Authors must better explain the statistical analysis applied to validate results, which kind of analysis and test were used. It is not sufficient to state: “All statistical analysis were performed with R and with the software Prism 8 (GraphPad) by using the default setting”.

I am little bit confused about the validation of the peptide array results. Both in Fig.1 and supplementary fig.1, it seems clear the higher fluorescence intensity (cumulative) coming from the convalescent vs. vaccinated sera. However, in Fig. 2A it is not reported the statistical significance between the two sera (only against control). Is it a real difference? Authors supposed that this difference was the result of a higher Ab titer of convalescent vs. vaccinated sera. Therefore, they measured the Ab titer against the spike protein of the two sera, finding the opposite result. Again, they did not report the statistical significance. If the difference is statistically valid, then the hypothesis explaining these results deserves a broader discussion.

Table 1 is unreadable in this form and must be improved, also indicating the number/name of the peptides. For example, on pag. 4 line 152, Authors refer to peptide 1, 2, and 4, however it is impossible to recognize these peptides in the table.

Minor points:

In all figures, panels must be indicated by uppercase letters

There are a few typing and grammatical errors: see for example on pag. 3 line 110; pag.4 line 145; pag. 9 line 178 and 179; etc. The use of comma is sometime inappropriate: see for example on pag “The regions of this helix, are enabled after dissociation of the 192 S1 domain and rearrangement of the structure into the post-fusion structure”

Author Response

Response to Reviewer #2:

Comment 1: The number of sera analyzed (31 convalescent, 25 vaccinated sera) is maybe limited for drawing definitive conclusions. It was indicated that the convalescent sera are from subjects who developed either severe or mild forms of the illness and that serum was collected at different time points. However, no indication of these conditions is reported in the further analysis. It would be interesting to identify, for example, the epitopes against which subjects with ARDS have rised antibodies compared to non-ARDS subjects. However, to this intent a higher number of samples should be analyzed.

Reply 1:. The main focus of this manuscript should is the analysis of the linear epitopes of vaccinated vs. convalescent individuals, as to our knowledge this point has not been investigated so far. An analysis of the linear epitopes recognized by sera of recovered individuals who have suffered from different severities of Covid-19 has been described already ( Zhang, Y.; Yang, Z.; et al., 2021. BMC Microbiol. 2021, 21, 194, doi:10.1186/s12866-021-02241). The data show clearly that more linear epitopes were detected by  sera from patients with ARDS.

In our study we wanted to detect only highly conserved epitopes. Therefore,  we chose a group of ARDS and non-ADRS patients. Nevertheless, analysis of our peptide array data in relation to the ARDS/non-ARDS groups could be interesting. As suggested by the reviewer, we analysed the data accordingly. As this analysis is based on a limited number of sera,  we present them in the supplement of the revised manuscript. In the text of the revised manuscript, we refer to this analysis but we mention the limitation of this analysis, since the sample size is in our opinion somewhat too small to be able to make concrete statements describing differences between ARDS and non-ARDS sera. (P8 L280ff)

Comment 2: Authors must better explain the statistical analysis applied to validate results, which kind of analysis and test were used. It is not sufficient to state: “All statistical analysis were performed with R and with the software Prism 8 (GraphPad) by using the default setting”.

Reply 2: As requested by the reviewer, we added more information about the statistical analysis. (P3 L127ff)

Comment 3: I am little bit confused about the validation of the peptide array results. Both in Fig.1 and supplementary fig.1, it seems clear the higher fluorescence intensity (cumulative) coming from the convalescent vs. vaccinated sera. However, in Fig. 2A it is not reported the statistical significance between the two sera (only against control). Is it a real difference? Authors supposed that this difference was the result of a higher Ab titer of convalescent vs. vaccinated sera. Therefore, they measured the Ab titer against the spike protein of the two sera, finding the opposite result. Again, they did not report the statistical significance. If the difference is statistically valid, then the hypothesis explaining these results deserves a broader discussion.

Reply 3: We added the missing statistics to Fig. 2 and changed the paragraph in the revised version of the manuscript. (P4 L157)

Comment 4: Table 1 is unreadable in this form and must be improved, also indicating the number/name of the peptides. For example, on pag. 4 line 152, Authors refer to peptide 1, 2, and 4, however it is impossible to recognize these peptides in the table.

Reply 4: We reformatted the table 1 and indicate all peptides in the Text with the respective peptide number from table 1. (P6 L220)

Comment 5: In all figures, panels must be indicated by uppercase letters

Reply 5: We changed all lowercase letters in the figures to uppercase.

Comment 6: There are a few typing and grammatical errors: see for example on pag. 3 line 110; pag.4 line 145; pag. 9 line 178 and 179; etc. The use of comma is sometime inappropriate: see for example on pag “The regions of this helix, are enabled after dissociation of the 192 S1 domain and rearrangement of the structure into the post-fusion structure”

Reply 6: We read the manuscript carefully and corrected the grammatical errors.

Round 2

Reviewer 1 Report

Thanks for answering all the questions and added the information according to the requirements. The quality of the manuscript has been significantly improved. However, some small mistake still founded in the article. Please correct these.

  1. Line 186 to 188:” 4 epitopes (S-5, S-12, S-18, S-37) are located in the transmembrane/cytoplasmic domain near the C-terminus and another 4 in the N-terminal domain (NTD).”

The 4 epitopes located in the transmembrane/cytoplasmic domain should be TM-242, TM-245, TM-246, CT-249. Another 4 in the N-terminal domain (NTD) should be (S-5, S-12, S-18, S-37) please correct it.

  1. Line 289: “ARDS patients with a Z score of =<1.5.” according to STable 1 it should be >=1.5.

Author Response

Point-by-point response

Date: 09.11.2021
Manuscript Number: vaccines-1447500
Title of Article: Comirnaty-elicited and convalescent sera recognize different spike epitopes
Name of the Corresponding Author: Eberhard Hildt
Email Address of the Corresponding Author: Eberhard.Hildt@pei.de

Response to Reviewer #1

Comment 1: Line 186 to 188:” 4 epitopes (S-5, S-12, S-18, S-37) are located in the transmembrane/cytoplasmic domain near the C-terminus and another 4 in the N-terminal domain (NTD).”

The 4 epitopes located in the transmembrane/cytoplasmic domain should be TM-242, TM-245, TM-246, CT-249. Another 4 in the N-terminal domain (NTD) should be (S-5, S-12, S-18, S-37) please correct it.

Reply 1: As requested by the reviewer, we We modified this paragraph and added the peptide numbers TM-242, TM-245, TM-246, CT-249 to the text.  (P5 L180)

Comment 2: Line 289: “ARDS patients with a Z score of =<1.5.” according to STable 1 it should be >=1.5.

Reply 2: We corrected this. (P8 L285)